# Early Cambrian fuxianhuiids from China reveal origin of the gnathobasic protopodite in euarthropods

Jie Yang[1], Javier Ortega-Hernández [2,3], David A. Legg [4], Tian Lan[5], Jin-bo Hou[1] & Xi-guang Zhang [1]

Euarthropods owe their evolutionary and ecological success to the morphological plasticity of their appendages. Although this variability is partly expressed in the specialization of the protopodite for a feeding function in the post-deutocerebral limbs, the origin of the former structure among Cambrian representatives remains uncertain. Here, we describe *Alacaris mirabilis* gen. et sp. nov. from the early Cambrian Xiaoshiba Lagerstätte in China, which reveals the proximal organization of fuxianhuiid appendages in exceptional detail. Proximally, the post-deutocerebral limbs possess an antero-posteriorly compressed protopodite with robust spines. The protopodite is attached to an endopod with more than a dozen podomeres, and an oval flap-shaped exopod. The gnathal edges of the protopodites form an axial food groove along the ventral side of the body, indicating a predatory/scavenging autecology. A cladistic analysis indicates that the fuxianhuiid protopodite represents the phylogenetically earliest occurrence of substantial proximal differentiation within stem-group Euarthropoda illuminating the origin of gnathobasic feeding.

[1] Key Laboratory for Palaeobiology, Yunnan University, Kunming 650091, China. [2] Department of Zoology, University of Cambridge, Downing Street, Cambridge CB2 3EJ, UK. [3] Department of Organismic and Evolutionary Biology and Museum of Comparative Zoology, Harvard University, 26 Oxford Street, Cambridge, MA 02138, USA. [4] Department of Earth, Atmospheric, and Environmental Sciences, University of Manchester, Manchester M13 9PL, UK. [5] College of Resources and Environmental Engineering, Guizhou University, Guiyang 550003, China. Correspondence and requests for materials should be addressed to X.-g.Z. (email: xgzhang@ynu.edu.cn)

Euarthropod appendages are characterized by the selective proximodistal (PD) differentiation of specific podomeres for various purposes, including locomotion, feeding, and sensing[1–4]. Despite its significance for the ecological success of these organisms, the early evolution of PD limb differentiation in the Cambrian stem lineage has only received minimal scrutiny. The origin of the protopodite—the specialized basal part of the post-deutocerebral limbs in crown-group euarthropods—is of critical importance in this context given its involvement in gnathobasic feeding[1,5–9]. Current hypotheses postulate that the appendicular organization of the crown-group evolved by the fusion of the proximal podomeres in the biramous limbs of Cambrian taxa, resulting in the simultaneous origin of the protopodite and the seven-segmented endopod observed in several representatives[5,10,11]. The fuxianhuiids—a group of stem-group euarthropods exclusively known from the Cambrian of South China—occupy a central role in these hypotheses, as their appendages have been regarded as an evolutionary link between the lobopods of soft-bodied ancestors[12] and the arthropodized limbs of Deuteropoda (i.e., upper stem-group Euarthropoda + crown-group Euarthropoda)[13]. Problematically, the basal portion of the fuxianhuiid post-deutocerebral appendages remains completely unknown despite numerous studies on the anatomy of these fossil organisms[14–19].

Here we describe the proximal functional morphology of post-deutocerebral appendages in three fuxianhuiid species based on exceptionally preserved fossil specimens from the early Cambrian (Stage 3) Xiaoshiba Lagerstätte in South China[12,14]. Our material includes a new taxon and provides direct evidence for the early evolution of the specialized protopodite with a feeding function within the euarthropod stem lineage.

## Results

### Systematic palaeontology.

(Upper stem-group) Euarthropoda Lankester 1904
(see discussion in ref. [13])
Fuxianhuiida Bousfield 1995
Chengjiangocarididae Hou and Bergström 1997

**Constituent taxa.** *Chengjiangocaris longiformis* Hou and Bergström, 1991[15] (Cambrian Stage 3, Chengjiang); *Chengjiangocaris kunmingensis* Yang et al., 2013[14] (Cambrian Stage 3, Xiaoshiba); *Alacaris mirabilis* gen. et sp. nov. (Cambrian Stage 3, Xiaoshiba). See also Supplementary Note 1.

*Alacaris mirabilis* gen. et sp. nov.

**Etymology.** After *Ala*, a village about 3.7 km northwest of the Xiaoshiba section; *caris* (Latin), shrimp; *mirab* (Latin), miracle, referring to the unexpected discovery of gnathal structures in fuxianhuiids limbs.

**Type material.** Yunnan Key Laboratory for Palaeobiology, Yunnan University; YKLP 12268 (holotype), a complete individual preserved in lateroventral view, showing details of the articulated appendages (Fig. 1a). YKLP 12269 (paratype), an articulated individual preserved in lateral view showing the complete organization of the dorsal exoskeleton and the ventral limbs (Fig. 1f).

**Other material.** 37 topotype specimens, YKLP 12270–12306.

**Locality and horizon.** Xiaoshiba section in Kunming, South China (Cambrian Series 2, Stage 3; local lower Canglangpuan Stage); lower portion of the Hongjingshao Formation typified by the presence of trilobite *Zhangshania*, corresponding to the *Yilangella–Zhangshania* biozone[20], approximately 10 m above the *Yunnanocephalus–Chengjiangaspis–Hongshiyanaspis* trilobite biozone[12] (Supplementary Fig. 1).

**Diagnosis.** Fuxianhuiid with heart-shaped head shield associated with eye-bearing anterior sclerite. Body composed of 13 overlapping tergites with pleurae that narrow posteriorly, plus a conical tailspine with lateral flukes. Five anteriormost tergites greatly reduced relative to the rest of the trunk, covered by head shield in life position. Appendages consist of elongate (deutocerebral) antennae, followed by a set of (tritocerebral) specialized post-antennal appendages (SPAs), and several pairs of biramous limbs. Four pairs of post-antennal limbs bear a robust protopodite with gnathobasic endites. Endopods consist of more than a dozen podomeres (i.e., multipodomerous), and a prominent terminal claw. Exopod flap like. Mouth opening and proximal bases of SPAs concealed by a broad hypostome with anterior margin extension, defined by lateral slits.

**Remarks.** *Alacaris* is similar to *Chengjiangocaris* in general body architecture, most notably the possession of a heart-shaped head shield, a broad butterfly-shaped hypostome, five anteriormost reduced trunk tergites, and other trunk tergites that taper in width posteriorly without expanded pleurae[14–17]. *Alacaris* differs from *Chengjiangocaris* in the presence of only 13 trunk tergites, the more robust construction of the SPAs with serrated edges, and less developed lateral wing-like extensions on the hypostome.

**Description.** Individuals reach 12 cm in maximum length. The dorsal exoskeletal morphology consists of a sub-elliptical anterior sclerite connected to a pair of lateral stalked eyes (Fig. 1g, h and Supplementary Fig. 2c–e). The anterior sclerite articulates with a broad heart-shaped head shield that covers the cephalic region and part of the trunk. The head shield is frequently taphonomically displaced[14], revealing details of the underlying anatomy (Fig. 1a, f, g and Supplementary Figs 2a, c and 3). The trunk comprises 13 tergites (T$n$) with a sub-rectangular outline (Fig. 1a, f and Supplementary Fig. 4c). T1 to T5 are greatly reduced, and together reach a sagittal length similar to T6 (Supplementary Figs 2a, f and 4a, c, d); these tergites are covered by the head shield in life position. T1 to T5 widen progressively posteriorly, and have anteriorly reflexed pleura with rounded margins. T6 to T13 maintain a sub-equal length throughout the body, but taper in width posteriorly (Fig. 1 and Supplementary Figs 2, 4); the pleurae have rounded anterior margins, and acute posterior ones. The tailspine has a conical shape, and is associated with a pair of tail flukes with posterior-facing setae (Fig. 1a and Supplementary Fig. 4c, e).

The ventral organization of the head includes a pair of deutocerebral[14,21] pre-oral antennae composed of approximately 20 articles that narrow distally (Fig. 1b, g, h and Supplementary Fig. 2c, d, g, h). The antennae attach close to the anterior margin of the hypostome, the sclerotized plate that covers the mouth (Fig. 1a–c, g, h and Supplementary Figs 2c, d and 3a, b). The hypostome has a broadly subtrapezoidal outline. The anterior margin extends medially, and is delimited by a pair of short slit-like lateral furrows. The lateral margins form wing-like extensions with straight edges. The posterior margin has a medial notch that conveys a bilobed appearance. The following limbs are a pair of tritocerebral[14,22] SPAs that occupy a para-oral position (Fig. 1a–c and Supplementary Fig. 2a, b, h, j), and whose proximal halves are covered by the hypostome. The SPAs consist of three robust sub-rectangular podomeres (P$n$). P1 is the shortest, and is distinguished by the presence of short spinose endites that form a medially oriented gnathal edge (Fig. 1b, c, and Supplementary Fig. 3a, b). P2 is the longest, and articulates with P3, which bears an acute distal termination and a strong spine on its posterior blade-like margin (Supplementary Fig. 2h–j). The surface of the SPAs is covered by coarse tubercles, which together with their

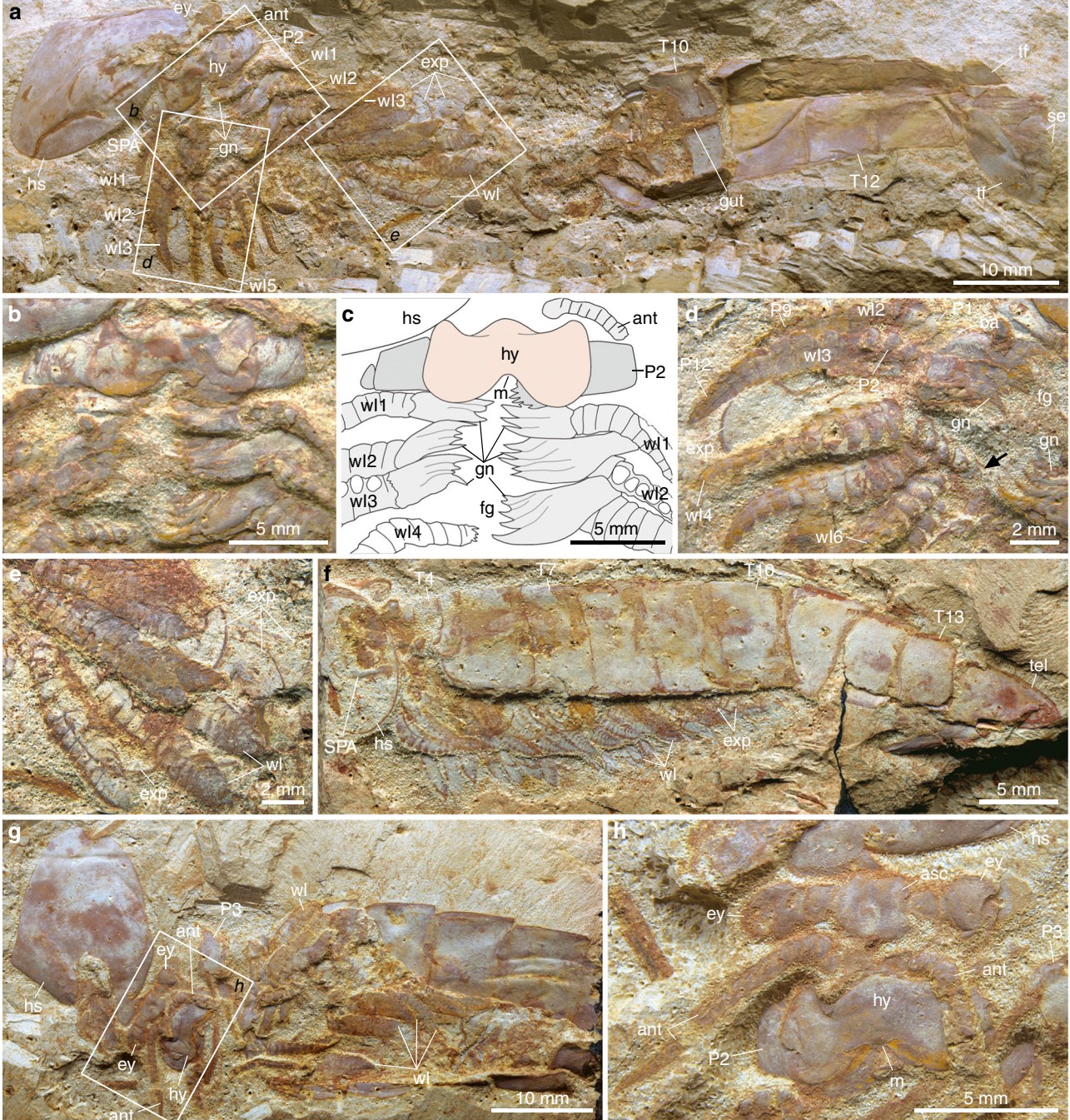

**Fig. 1** *Alacaris multinoda* from the Cambrian (Stage 3) Xiaoshiba Lagerstätte. **a–e** YKLP 12268 (holotype). **a** Complete individual in ventral view showing disarticulated head shield, appendicular organization, trunk tergites, and tailspine with paired flukes. **b** Details of area *b*, showing antennae, hypostome, SPAs, and three sets of walking legs with differentiated gnathobasic protopodites forming a ventral food groove. **c** Interpretative drawing of **b**. **d** Close-up of area *d*, showing multisegmented endopods with prominent protopodites, followed by walking legs with spinose endites (arrowed). **e** Close-up of area *e*, showing multisegmented endopods and flap-like exopods. **f** YKLP 12269, lateral view of a complete individual. **g** YKLP 12276, specimen with disarticulated head shield, showing organization of the anterior region. **h** Close-up of area *h*, showing the anterior sclerite with stalked eyes, the insertion of the paired antennae close to the anterior edge of the hypostome, and the proximal portions of the SPAs. ant: antenna, asc: anterior sclerite, exp: exopod, ey: eye, fg: food groove, gn: gnathobase, gut: alimentary canal, hs: head shield, hy: hypostome, m: mouth, P*n*: podomeres, se: setae, SPA: specialized post-antennal appendage, tel: tailspine, tf: tail fluke, T*n*: tergites, wl*n*: walking legs

three-dimensional preservation suggest a high degree of sclerotization.

The trunk contains numerous sets of biramous limbs that gradually decrease in size posteriorly (Fig. 1). T1 to T5 have a direct correlation with the underlying appendages, but the remaining tergites are associated with up to four limb pairs as indicated by leg impressions on the exoskeleton (Fig. 1f and Supplementary Fig. 2a). T11 to T13 lack appendages (Fig. 1a, f). The distal construction of the limbs is homonomous throughout the trunk, consisting of elongate endopods with up to 16

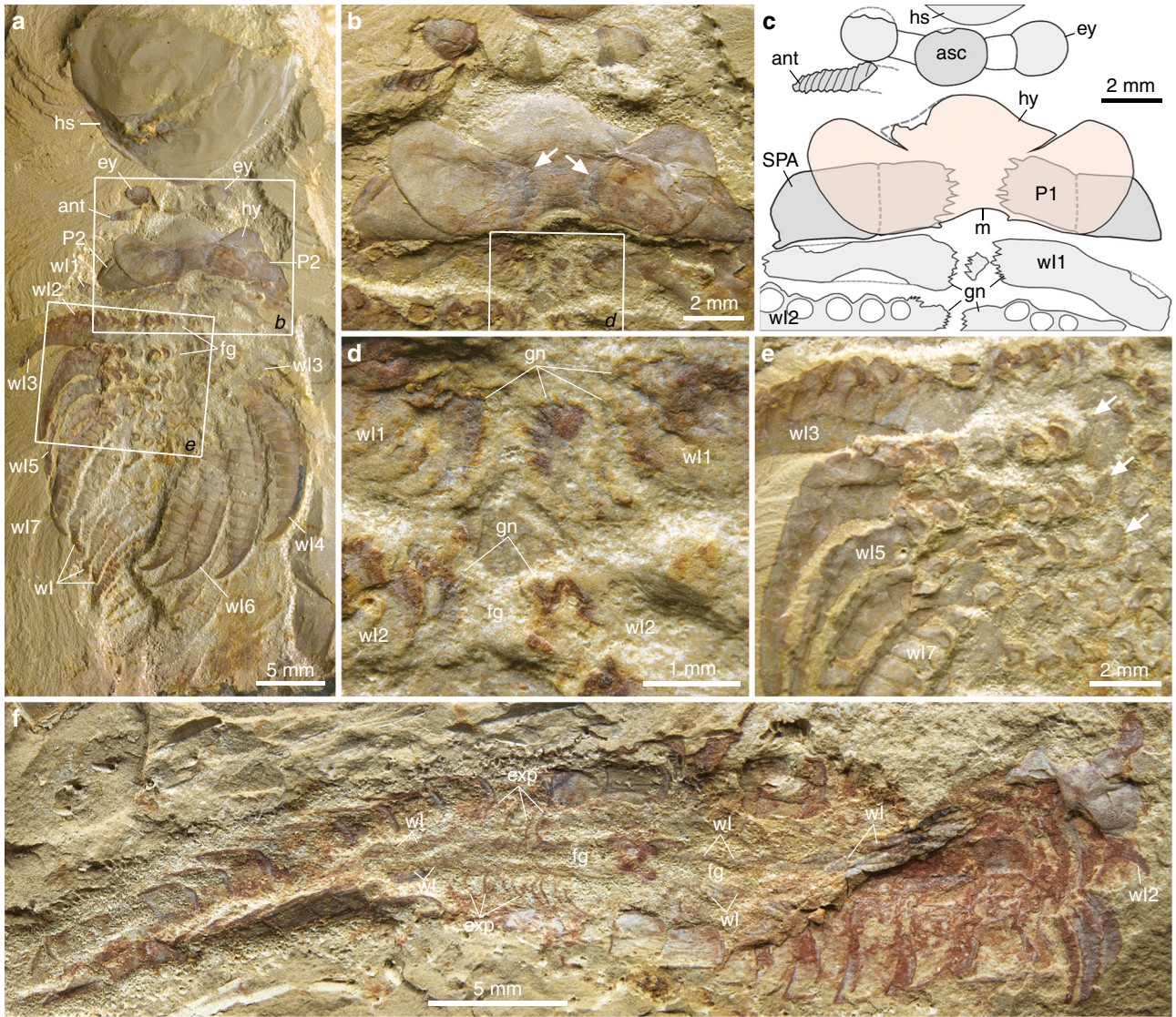

**Fig. 2** *Chengjiangocaris kunmingensis* from the Cambrian (Stage 3) Xiaoshiba Lagerstätte. **a** YKLP 12307, articulated specimen in ventral view. **b** Close-up of area *b*, showing the laterally expanded hypostome that overlaps the proximal portion of the SPAs, spinose endites (arrowed) of the SPAs are visible as impressions on the overlying hypostome, and the succeeding limbs also possess gnathobases. **c** Interpretative diagram of **b**. **d** Close-up of area *d*, showing gnathobases preserved on proximal portion of limbs located behind the SPAs, forming a ventral food groove. **e** Close-up of area *e*, showing the 3rd to 7th walking legs showing proximal spinose endites (arrowed). **f** YKLP 12308, ventral view of a nearly complete individual showing the food groove present between the two rows of closely spaced walking limbs. Abbreviations as in Fig. 1

podomeres that terminate in a conical distal claw (Fig. 1a, d and Supplementary Figs 3e and 4b), and an undivided flap-shaped exopod with short marginal setae (Fig. 1a, d–f and Supplementary Fig. 4b, d). Proximally, the appendages corresponding from T1 to T3 possess a prominent antero-posteriorly compressed protopodite, with up to five dorso-ventrally aligned robust spinose endites that project towards the body midline to form a ventral groove (Fig. 1a–d and Supplementary Figs 2a–c, 3a–d). The enlarged protopodite is absent in the remaining limbs, but these also possess medially directed spinose endites (Fig. 1a–d).

New collections also illuminate the ventral morphology of the Xiaoshiba fuxianhuiids *Chengjiangocaris kunmingensis* and *Fuxianhuia xiaoshibaensis*[14,17]. The entire in situ hypostome of *C. kunmingensis* is clearly observed for the first time (Fig. 2); it closely resembles that of *A. mirabilis* (compare Fig. 1a–c and Supplementary Fig. 2c, d), including the presence of a median anterior extension delimited by slit-like furrows. However, the

hypostome of *C. kunmingensis* has more developed wing-like lateral projections with rounded edges, the posterior notch is less prominent, and also flanked by a pair of short spines (Fig. 2a–c). The basal podomere of the robust SPAs carries gnathobasic edges with six spinose endites, visible as impressions through the overlying hypostome (Fig. 2b, c). At least three sets of biramous trunk limbs also have medially oriented gnathobasic edges that together form an axial ventral groove (Fig. 2d–f). Small endites are also preserved in the bases of the SPAs of *F. xiaoshibaensis* (Fig. 3e–g), although these are less prominent. The hypostome of *F. xiaoshibaensis* has a subtrapezoidal outline and bears an elevated medial region that appears to be aligned with the posterior-facing mouth opening (Fig. 3c–e); however, the posterior and lateral margins of the hypostome are broken and preclude further comparison with *Chengjiangocaris*. Neither *C. kunmingensis* nor *F. xiaoshibaensis* demonstrate the presence of enlarged protopodites as observed in *A. mirabilis* (Fig. 1a–c).

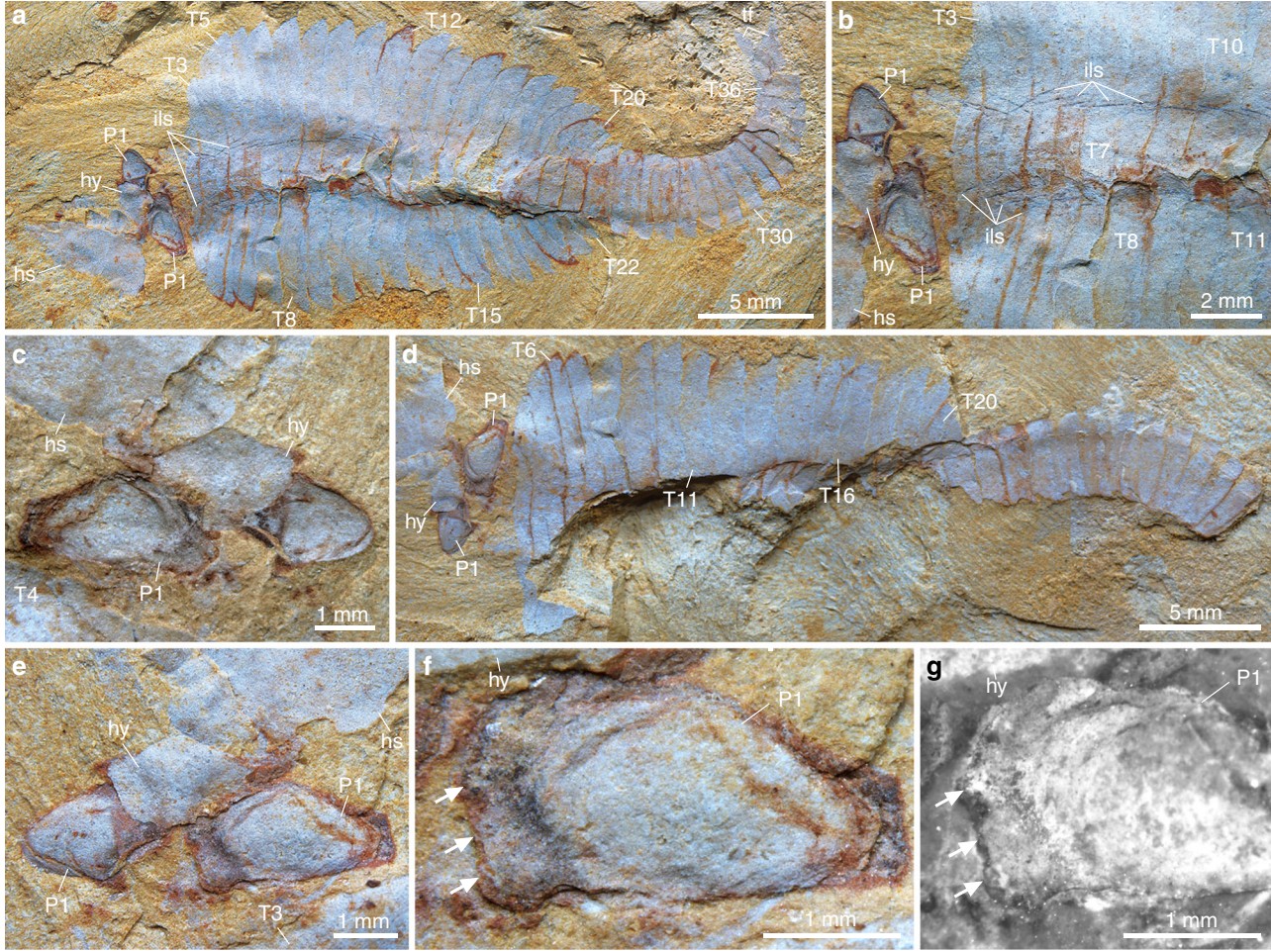

**Fig. 3** *Fuxianhuia xiaoshibaensis* from the Cambrian (Stage 3) Xiaoshiba Lagerstätte. **a–e** YKLP 12313a: **a** a complete articulated individual; **b** close-up of **a** showing regularly occurrence of leg sheath impressions; **c** close-up of the paired P1 of SPA covered by hypostome. **d–g** YKLP 12313b: **d** counterpart; **e** close-up of the paired P1 of SPA covered by hypostome; **f** details of **e** showing the left P1 with partially preserved endites situated along the inner margin (arrowed); **g** fluorescence image showing endites (arrowed). Abbreviations: as in Fig. 1

## Discussion

Xiaoshiba fossils provide clear evidence that the basal region of the post-deutocerebral limbs in fuxianhuiids is specialized for active gnathobasic feeding (Fig. 4). The presence of a differentiated protopodite with spinose endites indicates a predatory/scavenger autecology, in which prey would be processed by the gnathobases during the rhythmical movement of the legs, and transported along the ventral groove towards the posterior-facing mouth[1,6–8]. Food items closer to the mouth opening would be further manipulated by the interaction of the heavily sclerotized SPAs[14] and the broad posteriorly notched hypostome (Fig. 4c, d), which have a comparable organization to Palaeozoic euarthropods with a predatory/scavenger habitus[6–8,23,24]. These observations reject the prevailing view of fuxianhuiids as simple mud-feeders with exclusively locomotory trunk limbs[14,16,18,19], and instead add support to the interpretation that these stem-group euarthropods engaged, at least occasionally, in durophagous predation, as suggested by a single specimen preserved with a gut tract, which bears phosphatized midgut glands, containing fragments of eodiscoid trilobites from the Kaili biota[25]. These comparisons suggest that the feeding strategy of fuxianhuiids approximates that of the stem-group chelicerates *Sidneyia inexpectans*[6] and *Wisangocaris barbarhardyae*[26] in combining post-oral limbs with well-developed gnathobasic protopodites and evidence for shelly cololites. These observations

expand the scant Cambrian record of durophagy in total-group Euarthropoda[24], without necessarily ruling out the likelihood that fuxianhuiids also preyed or scavenged on soft-bodied food items. Thus, our material illuminates the autoecology of a major clade of stem lineage representatives, and demonstrates that protopodite-based gnathobasic feeding originated earlier within the evolutionary history of total-group Euarthropoda than previously considered[5,24,27,28].

A recent re-examination of the anterior organization of the Chengjiang radiodontan *Amplectobelua symbrachiata* has revealed paired ventral structures that possess strong spinose endites with a clear masticatory function[29], which could suggest an even earlier origin for gnathobasic feeding within lower stem-group Euarthropoda[13]. Despite their striking resemblance to the proximal limb morphology of trilobites[23] and *Limulus polyphemus*[29], and the jaws of onychophorans[30], it is uncertain whether the 'gnathobasic-like structures' of *A. symbrachiata* derive from the proximal or distal region of an appendage, which obscures their homology with the gnathobasic protopodite of more crown-wards representatives (Figs. 4 and 5). Given that the dorsal and ventral limb elements of radiodontans were not fused into a single biramous appendage[31], the 'gnathobasic-like structures' of *A. symbrachiata* cannot possibly derive from the protopodite in the traditional sense[3]. Instead, we regard the structures of *A. symbrachiata* as independently modified limbs for

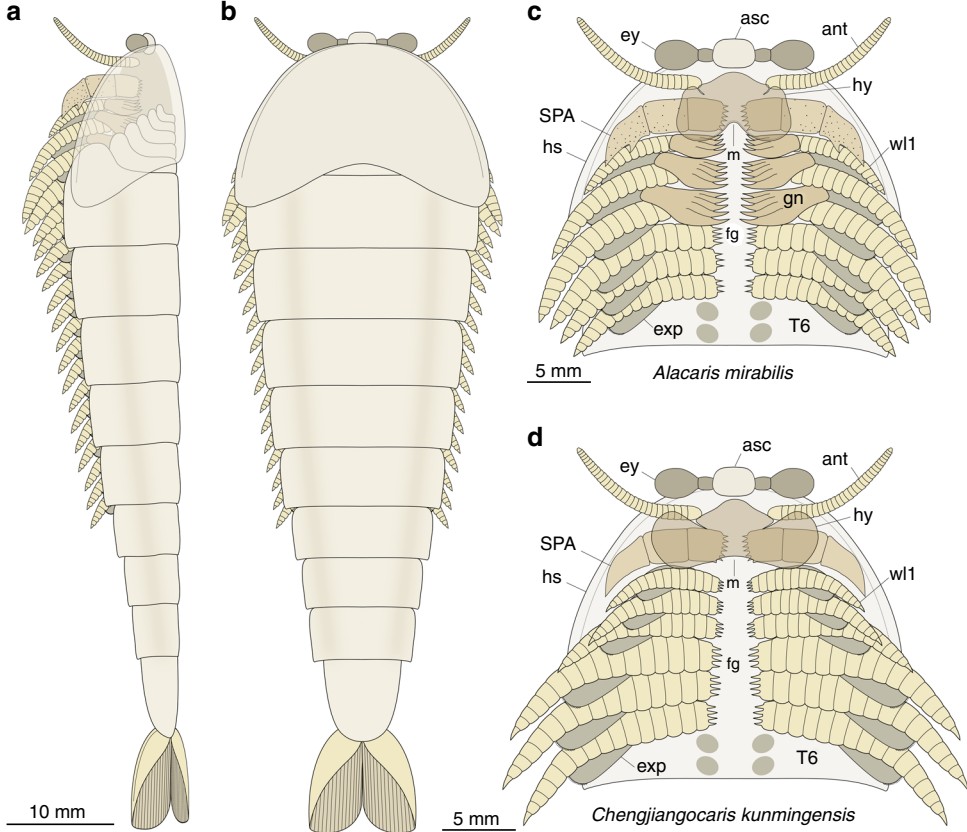

**Fig. 4** Morphological reconstruction of *Alacaris mirabilis* and *Chengjiangocaris kunmingensis*. **a** Full-body reconstruction of *A. mirabilis* in lateral view. **b** Full-body reconstruction of *A. mirabilis* in dorsal view. **c** Ventral view of the head and anterior trunk region of *A. mirabilis* showing gnathobasic protopodite ventral food groove. **d** Ventral view of the head and anterior trunk region of *C. kunmingensis*; note the differences in the morphology of the hypostome relative to *A. mirabilis,* and the presence of smaller gnathobasic endites on the proximal limbs. Abbreviations as in Fig. 1

food processing that worked in conjunction with the oral cone and the sclerotized frontal appendages, and which most likely originated through an analogous process of cephalization similar to that observed in other lineages within Panarthropoda[22].

The results of a phylogenetic analysis (Supplementary Fig. 5, Supplementary Note 1) resolve Fuxianhuiida as a monophyletic group within upper stem-group Euarthropoda[13] (Fig. 5a), placed a node above a paraphyletic grade of Cambrian bivalved forms (e.g., refs. [27,32,33]; contra [5,10,16,34]), and as sister group to a clade that includes megacheirans[28,35] and artiopodans[6,8,16,23]. These results inform the step-wise evolution of PD limb differentiation along the euarthropod stem lineage. The post-deutocerebral appendages of bivalved euarthropods—which comprise the most basal representatives of Deuteropoda[13,36]—consist of a series of flap-like exopod with marginal setae, and multipodomerous endopods; however, the structure of the proximal podomeres is generally obscured by the overlying carapace. In stem-wards forms, such as *Jugatacaris agilis*[32] and *Nereocaris* species[27,33], the slender endopods are composed of up to 30 homonomous podomeres with short endites that show no PD differentiation (Fig. 5b). More crown-wards representatives, like *Canadaspis perfecta*[37], show a slightly higher degree of limb specialization. The robust trunk endopods of *C. perfecta* consist of at least 11 freely articulating podomeres; those in the proximal half of the endopod bear endites with delicate comb-like terminations, whereas the endites on the seven distal-most podomeres are shorter and unbranched (Fig. 5b). However, the bivalved carapace of *C. perfecta* obscures the basal portion of the biramous appendages, and thus the organization of the protopodite is unknown (sensu ref. [37]; contra ref. [34]). A recent appraisal of the

trunk appendage morphology in the Burgess Shale bivalved euarthropods *Odaraia alata*[38], *Branchiocaris pretiosa*[39], and *Tokummia katalepsis*[34] suggests the presence of a segmented protopodite associated with an endopod with seven podomeres (see basipod in ref. [34]), which would imply a deeper origin for PD limb differentiation according to our topology (Fig. 5a; Supplementary Fig. 5). Evidence supporting the presence of a specialized protopodite in these taxa is contentious, however, as there is little structural PD distinction in the trunk appendages of these three taxa, other than the difference in width that can be expected along the length of the limbs[34,38,39]. The putative protopodite of *T. katalepsis* clearly displays complete transverse segmental boundaries that are morphologically indistinguishable from those defining the distal podomeres that constitute the endopod (see Figs. 1c, 2d, h, i, m, and extended data fig. 9h in ref. [34]). Although the putative protopodite of *T. katalepsis* is reconstructed as having more robust endites than the endopod, the figured fossil material demonstrates that the endopod podomeres possess similar endites as those observed in the proximal limb region, and thus show no signs of this proposed regionalization (compare figs. 1d and 3c in ref. [34]). The lack of substantial PD differentiation is more evident in *O. alata* (see extended fig. 9g, h in ref. [34]), as the limb portion regarded as the protopodite is again morphologically identical to the distal endopod—albeit slightly thicker—and also possesses complete transverse segmental boundaries. Finally, none of the figured material of *B. pretiosa* (see fig. 1i and extended data fig. 7e in ref. [34]) provides clear evidence of a differentiation between the proximal and distal portions of the trunk limbs. As such, the proximal limb organization of *O. alata*[38], *B. pretiosa*[39], and *T. katalepsis*[34] fundamentally differs from the

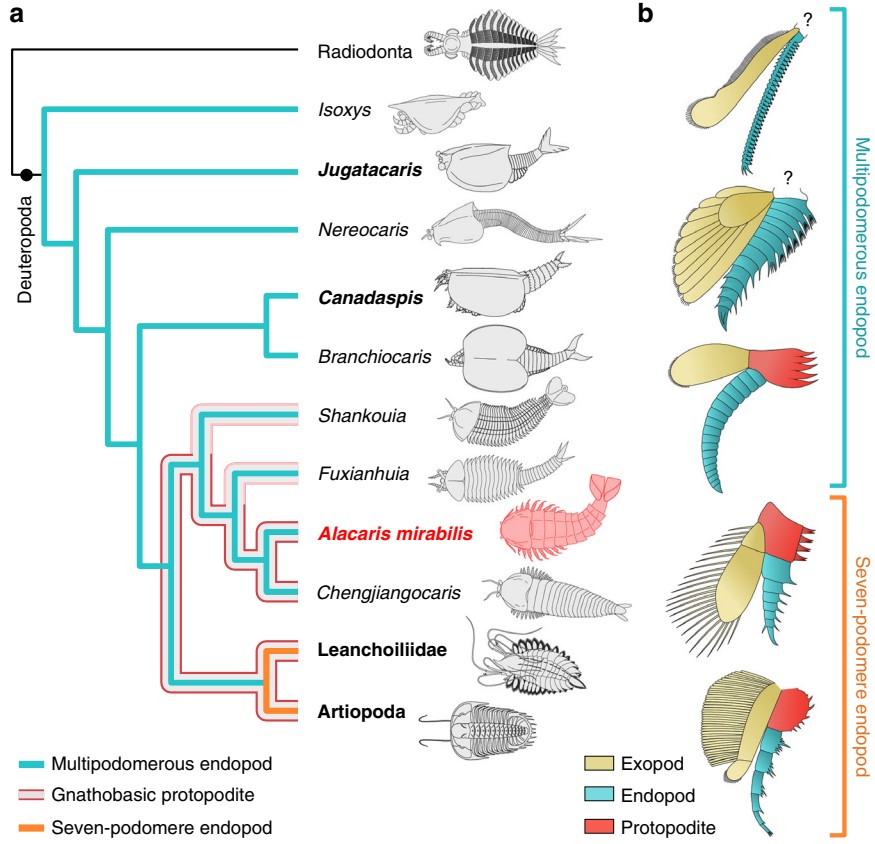

**Fig. 5** Evolution of euarthropod PD limb axis differentiation. **a** Simplified cladogram showing phylogenetic position of *Alacaris mirabilis*, origin of the gnathobasic protopodite, and evolutionary reduction in the number of endopod podomeres within stem-group Euarthropoda (detailed results presented in Supplementary Fig. 5b); the presence of gnathobasic protopodites on the trunk limbs of *Alacaris* and *Chengjiangocaris* suggests their possible occurrence in other fuxianhuiid taxa for which proximal limb data are not currently available. **b** Post-deutocerebral appendage organization and homology among stem and crown-group euarthropods. Limb reconstructions correspond to taxa highlighted in bold in **a**, ordered from top to bottom

morphologically and functional specialized compressed protopodite without segmental boundaries expressed in *A. mirabilis* (Figs. 1a–c, 4, 5).

The post-deutocerebral appendages of *A. mirabilis* represent the earliest occurrence of substantial PD differentiation within the euarthropod stem lineage, expressed in the discrete basal protopodite with up to five spinose endites combined with a multipodomerous distal endopod with a relatively simple construction (Figs. 4 and 5b). The presence of multipodomerous endopods is resolved as a symplesiomorphy of upper stem-group Euarthropoda[13], which argues against the interpretation of fuxianhuiid limbs as being structurally comparable relative to the those of lobopodians, which lack epidermal segmentation, sclerotized plates, flexible articulating membranes, and substantial PD differentiation (contra refs. [5,16,18,19]; see also ref. [40]). The analysis also indicates that the origin of the protopodite preceded the evolutionary reduction in the number of endopod podomeres (Fig. 5a), from more than a dozen in upper stem-group euarthropods[13,14,18,27,32,33,40] to seven (or fewer) in more crown-wards representatives such as megacheirans and artiopodans[3,5–8,16,23,28] (Fig. 5b). This result falsifies hypotheses in which the protopodite and the seven-podomere endopod evolved simultaneously from the fusion of the proximal podomeres in Cambrian stem-group representatives (sensu refs. [5,11]), and is better reconciled with gene expression data that suggest a possible developmental partition between the protopodite and the distal portions of the limb[2,3,9,41,42].

The differentiated protopodite constitutes an archetypical feature of the post-deutocerebral appendages in extinct taxa that

occupy a crown-wards position relative to fuxianhuiids[6–8,18,23,28,35] (Fig. 5a), as well as crown-group chelicerates[7,43] and mandibulates[3,4,44,45]. Thus, Xiaoshiba fuxianhuiids pinpoint the origin of the fundamental PD appendicular organization, and ancestral gnathobasic mode of feeding, that define the early evolution of the crown-group[1,3,5,40,45] as well as the foundations for the remarkable ecomorphological diversity of mouthparts observed in extant Euarthropoda[2,9,41].

## Methods

**Fossil repository and imaging**. Fossil material is deposited at the Key Laboratory for Palaeobiology, Yunnan University, Kunming, China. Specimens were photographed with a Nikon D3X fitted with a Nikon AF-S Micro Nikkor 105 mm lens, and a LEICA M205-C stereomicroscope fitted with a Leica DFC 500 digital camera with directional illumination provided by a LEICA LED5000 MCITM. Fluorescent photography was performed using a LEICA DFC 7000T monochrome digital camera (fitted with a LEICA 10450028 lens) attached to a LEICA M205 FA fluorescence stereomicroscope (green-orange fluorescence).

**Phylogenetic analysis**. The dataset (Supplementary Notes 1 and 2), consisting of 43 taxa and 84 characters, was analyzed using the freely available phylogenetic software TNT[46] (tree analysis using new technology) v.1.1. Analyses were performed using New Technology Search with Parsimony Ratchet[47], Sectorial Searches, Tree Drifting, and Tree Fusing[48], set to find the shortest tree 100 times. The analysis was first run under equal weights (Supplementary Fig. 5a), and a second iteration under implied weights with a concavity value of three (Supplementary Fig. 5b). See refs. [30] for a discussion on the applicability and utility of character weighting in phylogenetic analyses.

**Nomenclatural acts**. This published work and the nomenclatural acts it contains have been registered in ZooBank, the proposed online registration system for the International Code of Zoological Nomenclature (ICZN). The ZooBank LSIDs

Legend for **a**:
- Multipodomerous endopod
- Gnathobasic protopodite
- Seven-podomere endopod

Legend for **b**:
- Exopod
- Endopod
- Protopodite

(Life Science Identifiers) can be resolved and the associated information viewed through any standard web browser by appending the LSID to the prefix 'http://zoobank.org/'. The LSID for this publication is: urn:lsid:zoobank.org:act: E2B329EC-560B-463B-AC94-5E4F638F7A1A.

**Data availability**. The character dataset (Supplementary Notes 1 and 2) is available as supplementary information. Photographic material of the fossils is available from the authors upon request. Studied specimens are deposited at the Yunnan Key Laboratory for Palaeobiology, Yunnan University, Kunming.

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

## Acknowledgements
We thank K.-S. Du, J.-F. He, and K.-R. Li for assistance with fossil collection. This study was funded by the National Natural Science Foundation of China (U1402232, 41472022 and 41730318 to X.-G.Z. and J.Y.), Herchel-Smith Postdoctoral Fellowship and

Emmanuel College Bye-Fellowship, University of Cambridge (to J.O.-H.), and Department of Science and Technology, Yunnan Province (2015HA045 to X.-G.Z.).

## Author contributions

X.-g.Z. and J.Y. conceived the study. J.Y., T.L., and J.-b.H. collected the material. J.Y. prepared all specimens for photography. J.O.-H., and X.-g.Z. performed research, and wrote the manuscript with input from other authors. X.-g.Z. and J.O.-H prepared the figures. D.A.L. designed and performed the phylogenetic analysis. All authors discussed and approved the manuscript.

## Additional information

**Competing interests:** The authors declare no competing financial interests.

