## [Peer Review File · Nature Communications]

Reviewers' comments:

Reviewer #1 (Remarks to the Author):

General comments:

This is an interesting paper that describes a new Cambrian arthropod from the Xiaoshiba Biota of South China, and presents new morphological information on related (but previously described) taxa that also possess gnathobasic appendages. This discovery obviously has significant implications for the evolution of arthropod appendages (especially gnathobasic biramous appendages), which, as the authors note, is a longstanding issue in arthropod phylogenetics. Overall, the descriptions and characters for the cladistic analysis are accurate, and the illustrations are of a high standard and show the necessary details. I think the manuscript could potentially be published after revision, provided the authors consider the points below:

1. A rather pedantic comment about the name of the new taxon: While I am by no means a Latin scholar, I have been taught that it is best to use the same suffix (where possible) with regard to genus and species names. So, in this case, I would suggest *Alacaris multinodis*. However, my Latin references tell me that 'noda' actually means 'knob' or 'node', not 'segment'. The Latin word for 'segment' is, funnily enough, 'segmentum'. So I would suggest a name change to *Alacaris multisegmentis*.

2. Page 9, line 1: The new evidence presented in this manuscript regarding gnathobasic feeding in fuxianhuidids could be taken a bit further to briefly mention durophagy, which is supported by previous observations of a fuxianhuid-like arthropod from the Kaili Biota that has fragmentary trilobites preserved in its gut tract. As noted in a recent review article on Cambrian predation (Bicknell & Paterson 2017, *Biological Reviews*), convincing evidence for durophagy in Cambrian arthropods is very rare and is best demonstrated by taxa that possess gnathobasic appendages (e.g., *Sidneyia* and *Wisangocaris*), with some specimens exhibiting shelly cololites.

3. Page 9, lines 3-4: Here you claim that your findings demonstrate that gnathobasic feeding originated earlier within the evolutionary history of total-group Euarthropoda than previously considered. This would have certainly been the case as of only a few months ago, but the very recent paper by Cong et al. (2017, *BMC Evolutionary Biology*) on the revised morphology of *Amplectobelua* convincingly demonstrates that this (more basal) stem-group euarthropod possessed paired gnathobase-like structures. This is where you may need to be careful about how you define gnathobases, or more importantly, how structures identified as such map on to the euarthropod tree. Do you consider the gnathobase-like structures of *Amplectobelua* appendages to be homologous to the gnathobases of more crownward euarthropods? If so, then gnathobases (and gnathobasic feeding) would map on at the base of your tree in Figure 4. One problem is knowing whether the proximal portion of the gnathobasic appendages of *Amplectobelua* could be considered a protopodite, or even part

of a biramous appendage for that matter. Unfortunately, the evidence is inconclusive, despite the gnathobase-like structures being associated with body flaps. So what I think you're safe in saying is that you have earliest unequivocal evidence (phylogenetically speaking) for euarthropods that possess a biramous appendage with a gnathobasic protopodite, but you can't necessarily claim the origin of gnathobasic feeding – that title may go to Amplectobelua. In this regard, the title of your paper is rather misleading, so I would recommend modification.

4. Supp. fig. 5: In (d), you supposedly point to fine spines on the inner margin of the first podomere of the SPA. I simply cannot see them. Is it possible to enlarge this part to show the spines?

Reviewer #2 (Remarks to the Author):

General comments

The authors provide detailed key information on the proximal functional anatomy of the appendages of fuxianhuiids, an extinct group of arthropods of great evolutionary importance. Their findings are based on a sufficient number of fossil specimens distributed in three different species. Their interpretations are supported by strong fossil evidence. The results presented here have important bearings on the evolution and autecology of early arthropods (e.g. origin of gnathobasic feeding). The MS is well written and structured. Text-figures are excellent and very informative. I would strongly recommend this MS for publication in Nature Communications.

Page 2 line 21

Current hypotheses postulate that the appendicular organization of the crown-group evolved by the fusion of the proximal podomeres in the biramous limbs of Cambrian taxa, resulting in the simultaneous origin of the protopodite and the seven-segmented endopod observed in several representatives

Why a fusion of podomeres could result in the origin a protopodite and an endopod. It is not very clear to me.

Page 10 line 8-17

The protopodite of Tokummia seems to have more robust endites than the endopod, not only on the reconstruction but also on photographs. Tokummia and the fuxianhuiids presented here seems to have had comparable gnathobasic feeding. *O. alata*, *B. pretiosa*, and

T. katalepsis are not fundamentally different from XXXX in terms of feeding type. Perhaps more comments are needed here.

Page 11 line 1

It is clear that the appendages of fuxianhuiids are different from those of lobopodians

(please give examples of typical lobopodian appendages) but they still retain some of their characteristic features (multisegmented endopods). There is no reason to reject a transitional evolution. Please add more comments here. How would you define a lobopodian appendage ? (important for unspecialized readers)

Page 11 line 7

This result falsifies hypotheses in which the protopodite and the seven-podomere endopod evolved simultaneously from the fusion
This hypothesis is unclear (same as above).

Recently, Aria and Caron (2017 ; Nature) proposed a phylogeny of arthropods based on more than 80 taxa. Please explain briefly how your phylogeny differs from their results. Are you using the same cladistic methods ?

Reviewer #3 (Remarks to the Author):

Yang et al. present new data on different early Cambrian euarthropods, especially on their appendage morphology. As the appendages in arthropods are, from a functional-morphological point of view, the most important body parts, new information from the fossil record is always important.

The manuscript is in general well written and illustrated. Besides some minor comments mentioned below, I mainly see one important point that needs to be approached. The authors say that with their new findings, they can clearly show that these specialisations, especially concerning the feeding apparatus, occurred earlier along the lineage than previously estimated. However, this is only true if the taxa shown in Fig. 4 to bear gnathobasic protopodites all really bear them. For *Shankouia* I am not aware of gnathobasic protopodites, but this might be a secondary reduction (or a lack of knowledge from my side). However, also the photographs of *Fuxianhuia* presented in Suppl. Fig. 5 do not convince me that there are proper spines as mentioned by the authors. Therefore, I would suggest to show a larger magnification of Suppl. Fig. 5d, maybe also with a different contrasting method, and to put Suppl. Fig. 5 into the main text, as it is really important in this context.

Additionally, the authors need to discuss which implications it would have if neither *Shankouia* nor *Fuxianhuia* have gnathobasic protopodites. Independent evolution? Reduction? Different tree topology? I would prefer the latter option.

Therefore, I suggest publication after moderate revision.

Page 2, line 5: I am aware that the term 'upper stem-group' has been used by some of the authors already in earlier publications. However, there is still no clear characterisation provided where to draw the border between lower and upper stem-group etc. Hence, I

would rather avoid the term or explain more clearly which groups are included (also in other instances in the text).

Page 2, line 7: here it sounds as if the food groove formed by the gnathal edges would clearly point to a predator or scavenger; however, at least a predator also needs specialised appendages for grasping prey; rephrase

Page 3, line 8: I find it a bit weird to talk about 'exceptionally preserved fossil populations' as this is no study on population biology. 'exceptionally preserved fossil specimens' should be fine.

Page 3, lines 14-16: Is it journal style that there are no years included for the describers of the group names? If not, please add. Also, I would omit Linnean ranks, as they have no scientific value, and there is no rank for Euarthropoda used anyway.

Page 3, line 18ff.: Also here, I would add years to the describer names if the journal style allows it.

Page 4, line 4: 'noda' does not mean 'segment' in Latin, but 'knot' or 'node' (at least this is the major association one has when reading it); hence I would suggest to rephrase the explanation or change the name

Page 5, line 4: as 'antennae' is used for appendages of different segments in different arthropods, please give a clear reference which segment these appendages should belong to

Page 5, line 11: the possession of

Page 7, line 14: absent in the remaining

Page 9, line 2: 'clarify' is a rather unscientific word, use a more cautious expression

Page 9, line 11: 'representatives' instead of 'members'

Page 9, lines 11-12: That sounds as if these appendages only consist of endopod and exopod, rephrase

Page 9, line 13: Use complete species names, not just genus names

Figures (also in the supplement):

In general, the figures are nice, but the labelings are often hard to recognise on the photographs (white on bright background). I suggest to use a thin black stroke.

Additionally, I have some trouble to easily find out which marked area (marked with i, ii, iii) belongs to which panel. It would be easier to use the same letter as for the panel, but maybe in a different style.

Fig. 3: In comparison to the photographs, I think that in the reconstruction the antennae are fairly thin. Please check again.

Fig. 4: I find it rather disturbing that the reconstructed animals swim in different directions. I suggest to let them all face to the left.

Fig. 4: Please correct me, but I am not aware of any gnathobasic protopodites in Shankouia, but in the figure it is suggested that it has them. This might also point to a different phylogenetic position of Shankouia.

Suppl. Fig. 3: It is mentioned here that composite-fluorescence was used (typo: fluorescence, not flourescence). However, I did not find any description of these methods, only for normal illumination. Which wavelength was used, which filters? To me it looks like macrofluorescence, but obviously under sidelight conditions, which produces quite some artefacts (mainly shadows).

Suppl. Fig. 5d: The spines on podomere 1 are not visible to me, also not after higher magnification on my screen. To really show them (if present at all) a higher-magnified photograph would be necessary. As the presence (or absence) of gnathobasic protopodites in Fuxianhuia is crucial for the argumentation of the authors, this is very important. I would even strongly suggest to put this figure into the main text to have all data on gnathobasic protopodites in the main text.

Response to Reviewers

Reviewers' comments:

Reviewer #1 (Remarks to the Author):

General comments:

This is an interesting paper that describes a new Cambrian arthropod from the Xiaoshiba Biota of South China, and presents new morphological information on related (but previously described) taxa that also possess gnathobasic appendages. This discovery obviously has significant implications for the evolution of arthropod appendages (especially gnathobasicbiramous appendages), which, as the authors note, is a longstanding issue in arthropod phylogenetics. Overall, the descriptions and characters for the cladistic analysis are accurate, and the illustrations are of a high standard and show the necessary details. I think the manuscript could potentially be published after revision, provided the authors consider the points below:

1. A rather pedantic comment about the name of the new taxon: While I am by no means a Latin scholar, I have been taught that it is best to use the same suffix (where possible) with regard to genus and species names. So, in this case, I would suggest *Alacarismultinodis*. However, my Latin references tell me that 'noda' actually means 'knob' or 'node', not 'segment'. The Latin word for 'segment' is, funnily enough, 'segmentum'. So I would suggest a name change to *Alacarismultisegmentis*.

R. Following the reviewer's recommendation, we have renamed the new taxon as *Alacaris mirabilis*, mirab-, miracle, referring to the unexpected discovery of the gnathobasic feeding in fuxianhuiids.

2. Page 9, line 1: The new evidence presented in this manuscript regarding gnathobasic feeding in fuxianhuiids could be taken a bit further to briefly mention durophagy, which is supported by previous observations of a fuxianhuiid-like arthropod from the Kaili Biota that has fragmentary trilobites preserved in its gut tract. As noted in a recent review article on Cambrian predation (Bicknell & Paterson 2017, Biological Reviews), convincing evidence for durophagy in Cambrian arthropods is very rare and is best demonstrated by taxa that possess gnathobasic appendages (e.g., *Sidneyia* and *Wisangocaris*), with some specimens exhibiting shelly cololites.

R. Following the reviewer's recommendation, we have included a discussion of the significance of *Alacaris* for the evolution of durophagy in the euarthropod stem lineage and further elaborated on the evidence presented by Zhu et al. (2004) on the predation of agnostid trilobites by a putative fuxianhuiid from the middle Cambrian Kaili biota.

3. Page 9, lines 3-4: Here you claim that your findings demonstrate that gnathobasic feeding originated earlier within the evolutionary history of total-group Euarthropoda than previously considered. This would have certainly been the case as of only a few months ago, but the very recent paper by Cong et al. (2017, BMC Evolutionary Biology) on the revised morphology of *Amplectobelua* convincingly demonstrates that this (more basal) stem-group euarthropod possessed paired gnathobase-like structures. This is where you may need to be careful about how you define gnathobases, or more importantly, how structures identified as such map on to the euarthropod tree. Do you consider the gnathobase-like structures of *Amplectobelua* appendages to be homologous to the gnathobases of more crownward euarthropods? If so, then gnathobases (and gnathobasic feeding) would map on at the base of your tree in Figure 4. One problem is knowing whether the proximal

portion of the gnathobasic appendages of *Amplectobelua* could be considered a protopodite, or even part of a biramous appendage for that matter. Unfortunately, the evidence is inconclusive, despite the gnathobase-like structures being associated with body flaps. So what I think you're safe in saying is that you have earliest unequivocal evidence (phylogenetically speaking) for euarthropods that possess a biramous appendage with a gnathobasic protopodite, but you can't necessarily claim the origin of gnathobasic feeding – that title may go to *Amplectobelua*. In this regard, the title of your paper is rather misleading, so I would recommend modification.

R. We have slightly rephrased this sentence to clarify that we provide the earliest definitive evidence for gnathobasic feeding derived from the differentiated protopodite in euarthropods, as the gnathobasic-like structures observed in *Amplectobelua* cannot be regarded to originate from the protopodite, given that they predate the evolution of biramous appendages, as acknowledged by the reviewer and Cong et al. (2017). We further expand this discussion by clarifying this distinction for the reader and including reference to the work of Cong et al. (2017). Following the reviewer's suggestion, we have also slightly modified the title to include the clarification that our data reveals the origin of a gnathobasic protopodite in the evolution of euarthropods, which is directly linked with the origin of proximo-distal appendage differentiation in this clade.

4. Supp. fig. 5: In (d), you supposedly point to fine spines on the inner margin of the first podomere of the SPA. I simply cannot see them. Is it possible to enlarge this part to show the spines?

R. Change done.

Reviewer #2 (Remarks to the Author):

General comments

The authors provide detailed key information on the proximal functional anatomy of the appendages of fuxianhuiids, an extinct group of arthropods of great evolutionary importance. Their findings are based on a sufficient number of fossil specimens distributed in three different species. Their interpretations are supported by strong fossil evidence. The results presented here have important bearings on the evolution and autecology of early arthropods (e.g. origin of gnathobasic feeding). The MS is well written and structured. Text-figures are excellent and very informative. I would strongly recommend this MS for publication in Nature Communications.

Page 2 line 21

Current hypotheses postulate that the appendicular organization of the crown-group evolved by the fusion of the proximal podomeres in the biramous limbs of Cambrian taxa, resulting in the simultaneous origin of the protopodite and the seven-segmented endopod observed in several representatives

Why a fusion of podomeres could result in the origin a protopodite and an endopod. It is not very clear to me.

R. The hypothesis for the origin of the protopodite and seven-segmented endopod in crown-group euarthropods from the fusion of the proximal limbs of Cambrian taxa has been proposed, although not explained in great detail, mainly by Waloszek (2003), Waloszek et al. (2007) and Ito (1989). Waloszek has supported this view by hypothesizing an evolutionary sequence in which taxa with multipodomerous limbs, such as *Canadaspis*, undergo a fusion of the proximal part of the limb until achieving the condition observed in artiopodans. By contrast, Ito proposed the fusion of the proximal podomeres based on observations of the limbs in copepods, in which there are incomplete furrows that led him to hypothesize an ancestral state in which more free podomeres were part

of the endopod, but subsequently became integrated with the protopodite. As we clarify in the introduction and later in the discussion, neither of these hypotheses is well supported by other morphological or developmental data, and thus further highlights the contribution of the present study that the euarthropod protopodite originated in the proximal region of fuxianhuid limbs in association in a multipodomere distal endopod, rather than by fusion of the proximal podomeres themselves.

Page 10 line 8-17

The protopodite of *Tokummia* seems to have more robust endites than the endopod, not only on the reconstruction but also on photographs. *Tokummia* and the fuxianhuids presented here seems to have had comparable gnathobasic feeding. *O. alata*, *B. pretiosa*, and

T. katalepsis are not fundamentally different from XXXX in terms of feeding type. Perhaps more comments are needed here.

R. As we have discussed in this section, the proximal limb morphology of *Tokummia*, *Odaraia* and *Branchiocaris* offers no convincing evidence for substantial proximo-distal differentiation other than that expressed in the thickness of the appendage along its length, further complicated by the presence of clear transverse segmental boundaries on the proximal region of the limbs and the lack of clear evidence showing the attachment site of the exopods. These complications are best regarded as a case over-interpretation of the material of *Tokummia*, *Odaraia* and *Branchiocaris*, as the presence of a wider proximal region of the limbs is not only insufficient to demonstrate the presence of a podomere, but is actually expected given the approximately conical construction of euarthropod limbs in general. By contrast, the differentiated protopodite of *Alacaris* expressed a single robust structure that bears numerous spinose endites and which has no sign of internal segmental boundaries. We have rephrased this section in order to better communicate these points.

Page 11 line 1

It is clear that the appendages of fuxianhuids are different from those of lobopodians (please give examples of typical lobopodian appendages) but they still retain some of their characteristic features (multisegmented endopods). There is no reason to reject a transitional evolution. Please add more comments here. How would you define a lobopodian appendage? (important for unspecialized readers)

R. We have further elaborated some details on the construction of lobopodian limbs following the reviewer's suggestion. The reviewer is correct that there must have been an evolutionary transition from a lobopodous limb into an arthropodized limb, and thus we have removed this assertion in order to better communicate our conclusion. To clarify, it is not possible to argue that lobopodians have multisegmented endopods given that their limbs lack epidermal segmentation comparable to that of euarthropods.

Page 11 line 7

This result falsifies hypotheses in which the protopodite and the seven-podomere endopod evolved simultaneously from the fusion

This hypothesis is unclear (same as above).

R. Please see comments above.

Recently, Aria and Caron (2017; Nature) proposed a phylogeny of arthropods based on more than 80 taxa.

Please explain briefly how your phylogeny differs from their results. Are you using the same cladistic methods?

R. Similar to Aria and Caron (2017), we have also employed parsimony-based phylogenetic analyses to examine the new fossil data and the test the significance of our observations. However, we have designed a different and more manageable character database, given that we have major disagreements with the interpretation for some aspects of the morphology of Cambrian euarthropods employed by Aria and Caron, which generally reflect their hypothesis that bivalved forms unequivocally possess mandibles and other similar features in common with extant groups (e.g. see our criticism for the recognition of a differentiated protopodite in their fossil material). Thus, our dataset was designed to test these hypotheses using a broad sampling of fossil representatives and amore accurate character scoring that minimizes several assumptions of homology made by Aria and Caron.

Reviewer #3 (Remarks to the Author):

Yang et al. present new data on different early Cambrian euarthropods, especially on their appendage morphology. As the appendages in arthropods are, from a functional-morphological point of view, the most important body parts, new information from the fossil record is always important.

The manuscript is in general well written and illustrated. Besides some minor comments mentioned below, I mainly see one important point that needs to be approached. The authors say that with their new findings, they can clearly show that these specialisations, especially concerning the feeding apparatus, occurred earlier along the lineage than previously estimated. However, this is only true if the taxa shown in Fig. 4 to bear gnathobasicprotopodites all really bear them. For *Shankouia* I am not aware of gnathobasicprotopodites, but this might be a secondary reduction (or a lack of knowledge from my side). However, also the photographs of *Fuxianhuia* presented in Suppl. Fig. 5 do not convince me that there are proper spines as mentioned by the authors. Therefore, I would suggest to show a larger magnification of Suppl. Fig. 5d, maybe also with a different contrasting method, and to put Suppl. Fig. 5 into the main text, as it is really important in this context.

Additionally, the authors need to discuss which implications it would have if neither *Shankouia* nor *Fuxianhuia* have gnathobasicprotopodites. Independent evolution? Reduction? Different tree topology? I would prefer the latter option.

R. Following the reviewer's concern, we have modified Figure 5 (phylogenetic tree results) in order to clarify that gnathobasic protopodites on the trunk appendages have not been directly observed in *Shankouia* nor *Fuxianhuia*, but the presence of these structures in *Alacaris* and *Chengjiangocaris* suggest that these features may be expressed in other fuxianhuiids and simply not found yet due to insufficiently well-preserved material. We would prefer to refrain from making an explicit statement that similar feeding structures are authentically absent from *Shankouia*, *Liangwanshan*, *Fuxianhuia* and *Guangweicaris* given that only exceptionally preserved specimens of *Alacaris* and *Chengjiangocaris* in ventral view have demonstrated this aspect of the anatomy. We hope that future fossil discoveries will help to clarify the organization of the limbs in other fuxianhuiid taxa and thus further illuminate the diversity of feeding appendages in these stem-group euarthropods.

Therefore, I suggest publication after moderate revision.

Page 2, line 5: I am aware that the term 'upper stem-group' has been used by some of the authors already in earlier publications. However, there is still no clear characterisation provided where to draw the border between

lower and upper stem-group etc. Hence, I would rather avoid the term or explain more clearly which groups are included (also in other instances in the text).

R. Given that this comment applies to the abstract, we lack the space necessary to explain the distinction between lower and upper stem-group euarthropods, which has been addressed in detail by Ortega-Hernandez (2016 *Biological Reviews*). As elaborated on the latter publication, upper stem-group euarthropods are characterized by the presence of a multisegmented head with and deutocerebral first appendage pair, all of which are developmentally integrated as to provide a clear distinction from more stem-wards representatives such as radiodontans and lobopodians. Upper stem group euarthropods also possess other characters such as body arthropodization, limb arthropodization and biramous limbs, although these features were excluded from the definition of the group by Ortega-Hernandez (2016) in order to accommodate for future fossil discoveries that may inform these gaps in the evolution of euarthropods. References to Ortega-Hernandez (2016) are included in every mention of lower and upper stem group Euarthropoda for clarity.

Page 2, line 7: here it sounds as if the food groove formed by the gnathal edges would clearly point to a predator or scavenger; however, at least a predator also needs specialised appendages for grasping prey; rephrase

R. Please note that it is possible to be predator without the presence of grasping appendages, as demonstrated by similar adaptations in trilobites, other artiopodans and *Limulus*. Euarthropods feeding through gnathobasic limbs may scavenge as their primary feeding behavior, but are also able to prey on small or slow moving/sessile invertebrates (e.g. Zhu et al. 2004 *Biology Letters*; Zhai et al. 2016; Bicknell and Paterson 2017).

Page 3, line 8: I find it a bit weird to talk about 'exceptionally preserved fossil populations' as this is no study on population biology. 'exceptionally preserved fossil specimens' should be fine.

R. Change done.

Page 3, lines 14-16: Is it journal style that there are no years included for the describers of the group names? If not, please add. Also, I would omit Linnean ranks, as they have no scientific value, and there is no rank for Euarthropoda used anyway.

R. Change done in the main text and supplementary information files.

Page 3, line 18ff.: Also here, I would add years to the describer names if the journal style allows it.

R. Change done in the main text and supplementary information files.

Page 4, line 4: 'noda' does not mean 'segment' in Latin, but 'knot' or 'node' (at least this is the major association one has when reading it); hence I would suggest to rephrase the explanation or change the name

R. Change done. Please note the change in the name following the comments by Reviewer 1.

Page 5, line 4: as 'antennae' is used for appendages of different segments in different arthropods, please give a clear reference which segment these appendages should belong to

R. Change done.

Page 5, line 11: the possession of

R. Change done.

Page 7, line 14: absent in the remaining

R. Change done.

Page 9, line 2: 'clarify' is a rather unscientific word, use a more cautious expression

R. Change done.

Page 9, line 11: 'representatives' instead of 'members'

R. Change done.

Page 9, lines 11-12: That sounds as if these appendages only consist of endopod and exopod, rephrase

R. Change done.

Page 9, line 13: Use complete species names, not just genus names

R. Change done.

Figures (also in the supplement):

In general, the figures are nice, but the labelings are often hard to recognise on the photographs (white on bright background). I suggest to use a thin black stroke.

R. Change done.

Additionally, I have some trouble to easily find out which marked area (marked with i, ii, iii) belongs to which panel. It would be easier to use the same letter as for the panel, but maybe in a different style.

R. Change done.

Fig. 3: In comparison to the photographs, I think that in the reconstruction the antennae are fairly thin. Please check again.

R. Change done. Please note we have also added a reconstruction of *Chengjiangocaris kunmingensis* based on our new data to communicate our findings more clearly.

Fig. 4: I find it rather disturbing that the reconstructed animals swim in different directions. I suggest to let them all face to the left.

R. Change done.

Fig. 4: Please correct me, but I am not aware of any gnathobasicprotopodites in *Shankouia*, but in the figure it is suggested that it has them. This might also point to a different phylogenetic position of *Shankouia*.

R. Change done. We have modified the figure in order better communicate that gnathobasic structures are only directly observed in *Alacaris* and *Chengjiangocaris*, whereas those of *Shankouia* and have not been found, most likely due to the limited number of studies on the morphology of this taxon. However, the presence of gnathobasic protopodites in *Alacaris* and *Chengjiangocaris* suggests the possible presence of this character in other fuxianhuids.

Suppl. Fig. 3: It is mentioned here that composite-fluorescence was used (typo: fluorescence, not fluorescence). However, I did not find any description of these methods, only for normal illumination. Which wavelength was

used, which filters? To me it looks like macrofluorescence, but obviously under sidelight conditions, which produces quite some artefacts (mainly shadows).

R. Change done.

Suppl. Fig. 5d: The spines on podomere 1 are not visible to me, also not after higher magnification on my screen. To really show them (if present at all) a higher-magnified photograph would be necessary. As the presence (or absence) of gnathobasicprotopodites in Fuxianhuia is crucial for the argumentation of the authors, this is very important. I would even strongly suggest to put this figure into the main text to have all data on gnathobasicprotopodites in the main text.

R. Change done.

REVIEWERS' COMMENTS:

Reviewer #1 (Remarks to the Author):

I think the revised manuscript is a great improvement and has adequately addressed my original criticisms of the initial submission. I am happy to recommend publication, provided one very minor point is addressed:

* In lines 183-184, you state "...a single specimen preserved with phosphatized gut glands containing fragments of eodiscoid trilobites from the Kaili biota". This wording makes it sound like the fragmentary trilobites are preserved WITHIN the midgut glands, which clearly isn't the case. Perhaps you can rephrase by saying something like "...a single specimen preserved with a gut tract (and associated phosphatized midgut glands) containing fragments of eodiscoid trilobites from the Kaili biota".

Regards,
John Paterson

Reviewer #2 (Remarks to the Author):

The revised version of the MS looks very good to me.
No additional improvement is needed.

Reviewer #3 (Remarks to the Author):

The authors improved the manuscript significantly and it is, in my view, almost ready for acceptance. Yet, I found some minor errors, which still need to be corrected:

- Amplectobellua should be changed to Amplectobelua
- Though species names have been added to several of the genus names, they are still lacking in other cases. I found no species names for Sidneyia, Wisangocaris and Limulus, please check again and add.
- Some of the references in the list are lacking the year.
- In the caption of Fig. 1 the species name of the new species has not been changed yet.
- I still think that the contrast between the labelings and the photographs is fairly low in several images. I suggested to use a stroke, but leave this decision to the editor.

Response to reviewers' comments

Reviewer #1 (Remarks to the Author):

I think the revised manuscript is a great improvement and has adequately addressed my original criticisms of the initial submission. I am happy to recommend publication, provided one very minor point is addressed:

* In lines 183-184, you state "...a single specimen preserved with phosphatized gut glands containing fragments of eodiscoid trilobites from the Kaili biota". This wording makes it sound like the fragmentary trilobites are preserved WITHIN the midgut glands, which clearly isn't the case. Perhaps you can rephrase by saying something like "...a single specimen preserved with a gut tract (and associated phosphatized midgut glands) containing fragments of eodiscoid trilobites from the Kaili biota".

The sentence was rephrased as "...a single specimen preserved with a gut tract, which bears phosphatized midgut glands, containing fragments of eodiscoid trilobites from the Kaili biota" (p. 9, lines 2-4).

Regards,

John Paterson

Reviewer #2 (Remarks to the Author):

The revised version of the MS looks very good to me.

No additional improvement is needed.

Reviewer #3 (Remarks to the Author):

The authors improved the manuscript significantly and it is, in my view, almost ready for acceptance. Yet, I found some minor errors, which still need to be corrected:

- Amplectobellua should be changed to Amplectobelua

Corrected (p.9, line 14).

- Though species names have been added to several of the genus names, they are still lacking in other cases. I found no species names for Sidneyia, Wisangocaris and Limulus, please check again and add.

The missed species names have been added (p. 9, line 5, line 17).

- Some of the references in the list are lacking the year.

Publication years of three references have been added respectively (p. 16, line 18; p. 18, lines 17, 11).

- In the caption of Fig. 1 the species name of the new species has not been changed yet.

Changed.

- I still think that the contrast between the labelings and the photographs is fairly low in several images. I suggested to use a stroke, but leave this decision to the editor.

If checking the figures in the first version of the manuscript, one could find out that a few letters labeled in white have been replaced by black ones, or the background underneath a white letter has been carefully darkened without any remarkable damage to the illustration. In addition, of all figures a labeled letter is always regularly put in the upper right corner of each panel so that it's easy for readers to find out certain labeled panels without any confusion. In our view to make the labelings complex breaks the simplicity of the illustration.

** See Nature Research's author and referees' website at www.nature.com/authors for information about policies, services and author benefits

This email has been sent through the Springer Nature Tracking System NY-610A-NPG&MTS